

# Hydrological regime of Sahelian small water bodies from combined Sentinel-2 MSI and Sentinel-3 SRAL data

Mathilde de Fleury[1], Laurent Kergoat[1], and Manuela Grippa[1]

[1]Géosciences Environnement Toulouse (GET), UMR 5563, Université Toulouse 3, CNRS, IRD, 14 avenue Edouard Belin, OMP, 31400 Toulouse cedex 9, France

**Correspondence:** Mathilde de Fleury (mathilde.de-fleury@get.omp.eu), Laurent Kergoat (laurent.kergoat@get.omp.eu), Manuela Grippa (manuela.grippa@get.omp.eu)

**Abstract.** In the Sahelian semi-arid region, water resources, especially small water bodies such as ponds, small lakes and reservoirs in rural areas, are of vital importance. However, because of their high number and the scarce in situ monitoring networks, these resources and their spatio-temporal variability are poorly known at the regional scale. This study investigates the hydrological regime of 37 small water bodies, located in Mali, Niger and Burkina Faso, in Central Sahel. We propose a method based on remote sensing data only, which consists of combining water height data from Sentinel-3 SAR Radar Altimeter (SRAL) with water area data obtained with Sentinel-2 Multispectral Instrument (MSI) to create dense water height time series. Water height variations are then compared to evaporation estimated by the Penman–Monteith method using ERA5 reanalysis by the European Centre for Medium-Range Weather Forecasts (ECMWF) to infer water regimes during the dry season. Three main regimes stand out: a net water loss, mainly resulting from anthropogenic withdrawals, a net water supply occurring after the end of the rainy season through river network or water table exchange, and a balanced behaviour, where water losses during the dry season closely correspond to evaporation rates. Spatial patterns have been identified: in central Burkina Faso, most of the reservoirs show a net dry season water loss, which is explained by frequent irrigation, while reservoirs in northern Burkina Faso, generally show little water loss, indicating that water withdrawal is not significant in this area. Lakes located in the Inner Niger Delta in Mali and connected to the Niger river network generally show an important water supply, particularly at the beginning of the dry season. Lakes in Niger tend to show a weak signal toward water inflow that could be explained by exchange processes with the groundwater. These results show that satellite data are effective in estimating hydrological regimes as well as the anthropogenic impact on water resources, at the large scale, including resources found in small water bodies.

Key-words: small water bodies, hydrological regime, remote sensing, Sentinel-2, Sentinel-3, Central Sahel, Burkina Faso, Mali, Niger, evaporation, water loss, water supply.

## 1 Introduction

In the Sahel and more generally in West Africa, small water bodies are critical resources for the inhabitants, who use them on a daily basis to meet vital needs: drinking water, livestock watering, irrigation, fishing, and bathing among others (Cecchi et al., 2009; Frenken, 2005). These water bodies are widespread all over the region and include numerous small reservoirs, for



which dams have been built, small natural lakes and ponds, and intermediate situations where existing lakes are more or less developed. Burkina Faso, for example, built many reservoirs, whose number increased from about 200 in 1974 to about 1650 in 2008 (Cecchi et al., 2009). The aim of such actions was to address food security issues (Douxchamps, 2014) after severe droughts (Sally et al., 2011). The central Sahelian region also hosts a large number of small temporary water bodies (Haas et al., 2009; Gardelle et al., 2010; Papa et al., 2022) whose number is still not well known. There is a clear need to better survey

surface water resources in this region. Monitoring and understanding lake hydrological regimes is therefore an important step toward better management of these water resources.

      Since ground-based monitoring of water bodies in this area is usually restricted to some large lakes (mainly those supplying water to capital cities or used for electricity production), remote sensing data such as those provided by the Copernicus Sentinel

missions give an interesting tool to derive useful information. Radar altimetry monitors water heights (Birkett, 1994; Morris and Gill, 1994) by calculating the return time of a radar pulse emitted by the sensor onboard and reflected by the water surface. The Sentinel-3A and B satellites launched in 2016 and 2018 respectively carry a SAR Radar Altimeter (SRAL) on board. Their performance in measuring inland water levels has already been assessed, as in the Inner Niger Delta, resulting in an average Root Mean Square Error (RMSE) of 0.67 m (Normandin et al., 2018, Suppl. Table S5). The technology offered by Sentinel-3

provides a significant step forward with a much better resolved footprint than previous altimeters, allowing for the observation of smaller water bodies (Shu et al., 2020). Time series can be obtained from databases such as: DAHITI (Schwatke et al., 2015), G-REALM (Cooley et al., 2021) and HYDROWEB (Crétaux et al., 2011). Laser altimetry with ICESat-2 (Cooley et al., 2021) is also a technique currently used to measure water levels. Optical imagery is a powerful tool to detect surface water areas in cloud-free conditions, and recently several algorithms have been developed to map water bodies at the global scale (Pekel et al.,

2016; Messager et al., 2016; DeVries et al., 2017; Cordeiro et al, 2021). However, the conditions required by these algorithms are not always met in Central Sahel. This is due to the variability of water optical reflectances of these water bodies in time and space (e.g., Touré, 2016), caused by the common presence of aquatic vegetation (Gardelle et al., 2010), different levels of water turbidity, including extremely turbid and bright lakes (Robert et al., 2017), and by the seasonal variability of water body characteristics. The Modified Normalized Difference Water Index (MNDWI) is frequently used to differentiate water from soil

(Xu, 2006), usually with automatic or supervised thresholding methods. Using this index Reis et al. (2021) found that optimal thresholds still varied over time and space in the Sahel whereas Ji et al. (2009) showed that a fairly stable MNDWI threshold over time gives good results, even in the presence of mixed water and vegetation pixels.

      Studies combining surface water areas and heights estimated by remote sensing and/or a combination of remote sensing and

field measurements have been increasingly published during the last 5 to 10 years. Several works have been focused on large lakes. For example, Pham-Duc et al. (2020) developed a method based on remote sensing data to measure surface water extent and water volume variations in Lake Chad, the fourth largest lake in Africa. Sun et al. (2021) used remote and gauged data to estimate the water balance and water fluxes of Lake Poyang, China, over 20 years. Fewer studies focused on smaller water bodies in Europe and America (Baup et al., 2014; Schwatke et al., 2020; Gourgouletis et al., 2022). In Central Sahel, Gal et





al. (2016) estimated lake water inflow of Agoufou lake based on remote sensing data and evaporation modelling and validated the method with in situ measurements. Also in Central Sahel, Fowe et al. (2015) studied the water balance of a small reservoir in southern Burkina Faso, highlighting the variations caused by anthropogenic water withdrawal. Other studies assessed lakes topography through bathymetry (Arsen et al., 2013) or Digital Elevation Model (DEM, Avisse et al., 2017) to retrieve lakes storage. The variability of reservoirs at the global scale has been addressed by some recent works. For example, Cooley et al.

(2021) showed the great seasonal variability of reservoirs worldwide, drawing attention to the anthropogenic impacts on water resources, and Hou et al. (2022) highlighted the important role of precipitation in the observed variabilities. However, these global studies do not include a precise quantification of water fluxes over small water bodies in the Sahel, and several questions remain unanswered: What is the hydrological regime of these small water bodies? What are the dominant water exchanges in this region? How can their contribution be quantified? Is there a major anthropogenic impact on these water resources?

This work develops a methodology based on remote sensing data to quantify the hydrological regime of small water bodies in Central Sahel and derives information about their seasonal and interannual variability. It allows us to better understand the major processes at play in this region and identify human impact on these water resources.

## 2  Materials

### 2.1  Study site and lake selection

The study area is located in Central Sahel and includes water bodies in Mali, Burkina Faso and Niger. It covers arid, semi-arid (Sahelian) and sub-humid (Soudanian) areas according to Andam-Akorful et al. (2017) classification, with well-defined rainy and dry seasons enforced by a tropical monsoon system. The rainy season starts in June and ends in October, with variations due to latitude (Frappart et al., 2009; Panthou et al., 2018). The North's rainy season is shorter and annual rainfall ranges from around 200 mm.yr$^{-1}$ to 900 mm.yr$^{-1}$ from the North to South area. Four regions of interest can be defined based on different

geomorphology and development policies: the Inner Niger Delta, the centre of Burkina Faso around the capital Ouagadougou, which is densely populated and has a large number of reservoirs, the western area of Niger, and the Burkina Faso northern borders with Mali and Niger. Numerous water bodies are located in the study area, but the Sentinel-3 satellite orbits constrain selection of water bodies. The inter-track distance of 104 km for one satellite and 52 km for a combination of the two (Fig. 1), with a footprint of 300 m below the track, reduces the observable surface. Due to a potential track shifting of $\pm$ 1 km at

maximum (Crétaux et al., 2018) lakes located between 0 and 0.3 km from the nominate altimeter track have been included in the potential lakes to be studied. Using the maximum water extent of the Global Surface Water dataset (Pekel et al., 2016), 150 lakes were detected below the tracks. Among them, 42 had suitable altimeter data to provide long and consistent time series for the analysis. This amounts to 26.2 % of the lakes initially detected (Fig. 1 and Table 1): 21 are located in Burkina Faso, including 19 reservoirs, 12 in Mali and 9 in Niger, including 1 reservoir.





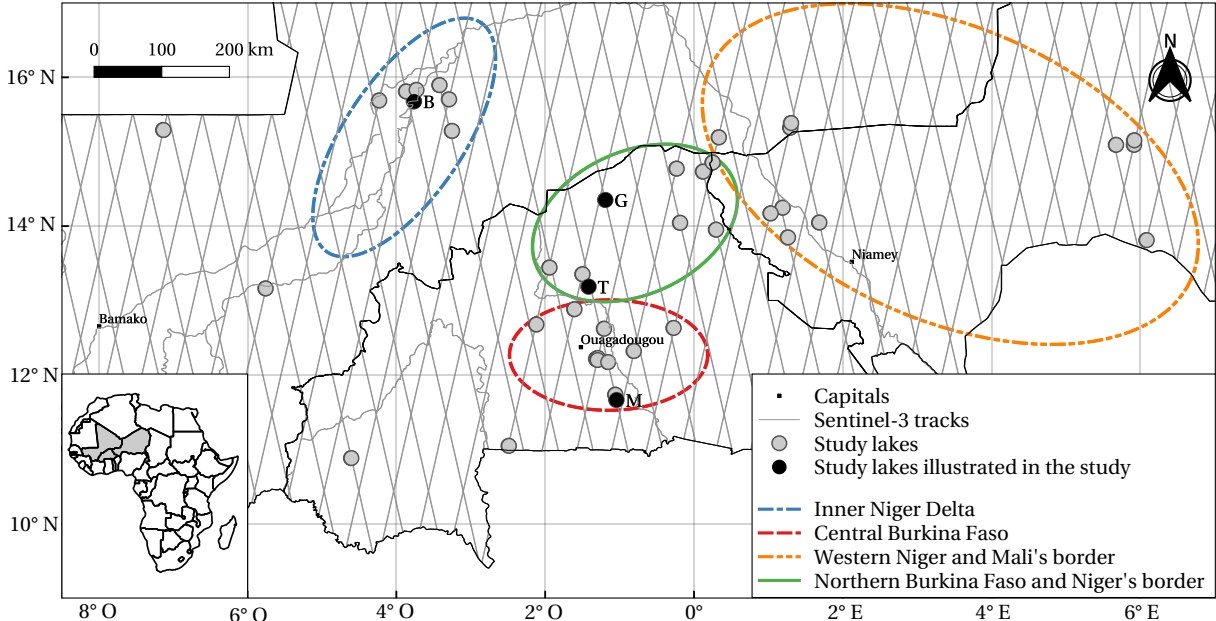

**Figure 1.** Study site and lakes in Central Sahel (Mali, Burkina Faso and Niger).

## 2.2 Data

Water areas and water albedos are derived from the freely available Sentinel-2 (S2) Multispectral instrument (MSI) images using the top-of-atmosphere reflectance products from the dataset "Sentinel-2 MSI: MultiSpectral Instrument, Level-1C", provided by Google Earth Engine (Gorelick et al., 2017). Measurements are made in 13 optical bands from VNIR to SWIR at a resolution of 10, 20 or 60 m depending on the band. In this study we used the blue band (B2) at 0.490 μm, the green band (B3) at 0.560 μm, the red band (B4) at 0.665 μm, all at 10 m resolution, and two SWIR bands (B11 and B12) at 1.610 and 2.190 μm with a 20 m resolution. Images are acquired from 2015 to present, with a revisit frequency of 10 days before the launch of Sentinel-2B in 2017 and five days afterwards, which allows good temporal monitoring, except when it is cloudy. Water heights are obtained from Sentinel-3 SRAL data (S3), provided by the Centre de Topographie des Océans et de l'Hydrosphère (CTOH, Frappart et al., 2021), referenced with a EGM2008 geoid model. The temporal frequency of measurements is 27 days, with a spatial resolution of 300 m (along-track) × 1.64 km (across-track), from 2016 to present. Precipitation is estimated by the Integrated Multi-satEllite Retrievals algorithm of the international satellite mission Global Precipitation Measurement (IMERG-GPM, Huffman et al., 2019). The data are provided by Google Earth Engine, through the "GPM: Global Precipitation Measurement (GPM) v6" collection, with a spatial resolution of $0.1° × 0.1°$ and a temporal resolution of 30 min. Other meteorological data are provided by the "ERA5 reanalysis hourly data on single levels from 1959 to present" database (Hersbach et al., 2018) produced by the European Centre for Medium-Range Weather Forecasts (ECMWF) within the Copernicus Climate Change Service (C3S). The data are provided at the resolution $0.25° × 0.25°$. Evaporation rate data provided by the





Global Lake Evaporation Volume (GLEV) dataset (Zhao et al., 2022) and Colorado pan evaporation data over a small reservoir (Boura) from Fowe et al. (2015), are used for validation over the April 2012–April 2014 period.

## 3    Method

### 3.1    Lake water balance estimation

The water balance approach defines the different fluxes controlling a water body regime (Winter, 1995). In this paper, we adapt the equation developed by Fowe et al. (2015), who propose a water balance equation expressed as the variations of water volumes applied to a small reservoir in Burkina Faso (Boura). All terms can be also expressed in water height, which fits our data better. The water height variation between two dates $t_0$ and $t_1$ can be written as the sum of precipitation and evaporation over the same period minus a residual term, referred to as "residual water balance" (R). This term is the net results of different hydrological fluxes: water inflow from the watershed into the lake, groundwater inflow, water losses due to overflow, infiltration losses, and withdrawals due to anthropogenic uses and can be expressed in millimetre per day as:

$$R = \frac{1}{t_1 - t_0} \left[ \Delta H_{t_0, t_1} - \sum_{i=t_0}^{t_1} (P_i + E_i) \right]$$ (1)

where $P_i$ (mm) and $E_i$ (mm) respectively indicate the daily precipitation over the lake and the daily evaporation from the lake. During the dry season, precipitation ($P_i$) is null. Evaporation ($E_i$) is estimated using available meteorological data. Water height variation ($\Delta H_{t_0, t_1}$) is estimated using altimetric data. As the altimeter offers data with a temporal resolution of 27 days, the time series are completed by other water heights estimations derived from water areas through an area-height relationship, called A-H curve, or sometimes hypsometric curve.

### 3.2    Lake water height estimation

Water height time series are extracted through the Altimetric Time Series Software (AlTiS, version 2.0, Frappart et al., 2021), which is an open-source software developed by CTOH. The process of extracting the time series is partially manual and has to be done for each lake. Among the Geophysical Data Record (GDR) variables proposed, the backscatter coefficient having a very high value for surface water is appropriate to distinguish water from soil (Taburet et al., 2020). The backscatter coefficient is extracted for data within the lake polygon. Samples that do not correspond to water are removed by thresholding. Following an empirical analysis, a threshold of 40 dB is retained. This is in line with Kittel et al. (2021) and Taburet et al. (2020) who propose thresholds of 30 and 45 dB respectively. Kittel et al. (2021) also state that changes in data processed from 2020 onwards result in a drop in the backscatter coefficient of 18 dB: the threshold has been therefore set to 22 dB for acquisition dates posterior to 2020. To reduce the influence of remaining outliers on the resulting time series, the median values are extracted (Fig. 2) as suggested by Frappart et al. (2021) and a threshold of 0.25 m is applied on the associated median absolute deviations (MAD). Water-like echoes created by wet sand, that may appear when the water body is empty, may also lead to





**Table 1.** Information and key results on study lakes. Lakes with a bold label are also shown in Sect. 4. Country abbreviations are: BF for Burkina Faso, M for Mali, N for Niger. The "nan" values indicates that conditions for five-year average residual water balance calculation have not been met.

| Lake label[a] | Country | Coordinates | MNDWI threshold | Average albedo | Average water area (km$^2$) | Average water height variation[b] (m) | Five-year average residual water balance (mm.d$^{-1}$) |
|---|---|---|---|---|---|---|---|
| Arzuma | BF | 12.218,–1.298 | –0.10 | 0.11 | 2.54 | 2.67 | –3.11 |
| B1 | BF | 12.171,–1.160 | 0.20 | 0.12 | 0.13 | 2.27 | –10.11 |
| Babou | BF | 12.882,–1.613 | –0.20 | 0.17 | 0.43 | 1.74 | nan |
| **Bakafé** | M | 15.678,–3.771 | –0.05 | 0.13 | 2.34 | 1.51 | 1.63 |
| Bam | BF | 13.353,–1.504 | –0.20 | 0.11 | 22.31 | 1.69 | nan |
| Barkea | BF | 14.044,–0.191 | 0.00 | 0.15 | 10.89 | 2.18 | –1.30 |
| Bokoko | M | 15.891,–3.431 | 0.00 | 0.15 | 0.39 | 1.59 | 2.79 |
| Boura | BF | 11.041,–2.486 | –0.25 | 0.10 | 1.44 | 2.95 | –1.20 |
| Dyaloub | M | 15.292,–7.137 | –0.30 | 0.15 | 5.73 | 0.89 | nan |
| Galigel | M | 15.194,0.324 | –0.10 | 0.13 | 2.25 | 2.38 | 0.16 |
| Gidan | N | 13.813,6.082 | 0.20 | 0.17 | 1.29 | 0.90 | 0.86 |
| **Gomde** | BF | 14.351,–1.197 | 0.00 | 0.19 | 28.22 | 1.71 | –1.54 |
| Hagoundou | M | 15.716,–3.295 | 0.00 | 0.09 | 37.91 | 1.13 | 9.71 |
| Iribakat | N | 15.091,5.910 | 0.00 | 0.14 | 0.27 | 2.00 | –1.10 |
| Kaboukoga | N | 14.047,1.678 | 0.10 | 0.22 | 0.24 | 1.21 | –0.09 |
| Koankin | BF | 11.733,–1.063 | 0.00 | 0.18 | 0.02 | 1.60 | nan |
| Korarou | M | 15.280,–3.258 | –0.30 | 0.13 | 36.27 | 1.33 | 2.46 |
| Kormou | M | 15.689,–4.237 | –0.20 | 0.13 | 3.65 | 2.80 | –2.29 |
| Koumaira | M | 15.810,–3.874 | –0.20 | 0.11 | 1.22 | 1.74 | 2.96 |
| M3 | M | 15.383,1.303 | 0.10 | 0.13 | 0.30 | 2.33 | 0.82 |
| M42 | M | 15.323,1.289 | –0.10 | 0.17 | 0.56 | 2.12 | 0.58 |
| **Manga** | BF | 11.663,–1.047 | –0.15 | 0.11 | 0.48 | 1.96 | –8.28 |
| Mogtedo | BF | 12.332,–0.804 | 0.20 | 0.13 | 2.36 | 1.54 | –6.49 |
| N10 | N | 15.149,5.924 | 0.00 | 0.11 | 0.55 | 2.63 | 0.19 |
| N4 | N | 14.246,1.165 | 0.00 | 0.20 | 4.98 | 1.74 | 0.00 |
| Nabitenga | BF | 12.618,–1.213 | –0.20 | 0.13 | 0.74 | 2.97 | –11.87 |
| Nazounga | BF | 12.675,–2.126 | –0.20 | 0.16 | 0.15 | 1.89 | –1.93 |
| Northern Tanvi | BF | 12.230,–1.303 | 0.00 | 0.11 | 0.13 | 3.09 | –12.45 |
| OuroDaka | BF | 13.955,0.297 | 0.00 | 0.15 | 3.13 | 2.02 | –1.48 |

[a]Labels can be defined by: Cecchi (2014) nomenclature or nearby villages or a letter representing the country associated with a number.

[b]Calculated from seasonal variations.





| Lake label[a] | Country | Coordinates | MNDWI threshold | Average albedo | Average water area (km$^2$) | Average water height variation[b] (m) | Five-year average residual water balance (mm.d$^{-1}$) |
|---|---|---|---|---|---|---|---|
| Seguenega | BF | 13.441,–1.952 | –0.20 | 0.13 | 1.44 | 2.00 | –2.18 |
| Southern Tanvi | BF | 12.199,–1.302 | 0.20 | 0.18 | 0.22 | 1.98 | –3.42 |
| Tabalakh | N | 15.063,5.651 | –0.10 | 0.14 | 6.98 | 1.90 | –0.39 |
| Tambao | BF | 14.733,0.117 | 0.05 | 0.13 | 0.15 | 1.39 | –2.50 |
| Tamou | N | 13.848,1.257 | 0.00 | 0.21 | 0.04 | 1.50 | –3.06 |
| **Tibin** | BF | 13.163,–1.391 | 0.10 | 0.14 | 15.99 | 1.93 | –1.35 |
| Timba | M | 15.832,–3.735 | –0.20 | 0.12 | 2.62 | 3.23 | –10.85 |
| Toussiana | BF | 10.880,–4.612 | 0.00 | 0.10 | 1.26 | 2.85 | –12.04 |
| Yakouta | BF | 14.772,–0.242 | –0.10 | 0.17 | 0.28 | 1.25 | –1.60 |
| Yaongo | BF | 12.619,–0.271 | –0.10 | 0.17 | 0.95 | 1.97 | –3.26 |
| Yumban | N | 14.861,0.246 | 0.00 | 0.18 | 19.28 | 2.15 | –6.57 |
| Zandela | M | 13.162,–5.757 | 0.00 | 0.12 | 0.60 | 1.76 | –1.81 |
| Zoribi | N | 14.170,1.021 | –0.25 | 0.17 | 1.26 | 0.98 | nan |

[a]Labels can be defined by: Cecchi (2014) nomenclature or nearby villages or a letter representing the country associated with a number.

[b]Calculated from seasonal variations.

outliers. To best prevent the use of such erroneous height values, data corresponding to periods when the water area is zero are removed.

## 3.3 Surface water area estimation

Sentinel-2 water optical reflectance ($\rho$) are firstly pre-processed to mask clouds using the Sentinel-2 QA band at 60 m (QA60) and an additional blue band threshold so that only values with $\rho_{B2} < 0.2$ are retained. Water detection (Fig. 2) is performed by applying a thresholding on the Modified Normalized Difference Water Index (MNDWI, Xu, 2006):

$$\text{MNDWI} = \frac{\rho_{\text{SWIR1}} - \rho_{\text{green}}}{\rho_{\text{SWIR1}} + \rho_{\text{green}}} \tag{2}$$

The MNDWI threshold (Table 1) is chosen ad hoc for each lake and kept constant over the study period. Highly negative thresholds are mostly used for lakes with vegetation.

## 3.4 Area-height curve estimation and water height time series densification

To estimate the A-H curve (Fig. 3), water heights and water areas are combined. We select quasi-simultaneous data in a $\pm$ 3 days interval, which follows Gao et al. (2012) work with MODIS and altimetry data. Based on Crétaux et al. (2016), a two degree polynomial curve is fitted to the data. Data outside the 95 % prediction interval (area within the dotted lines in Fig.

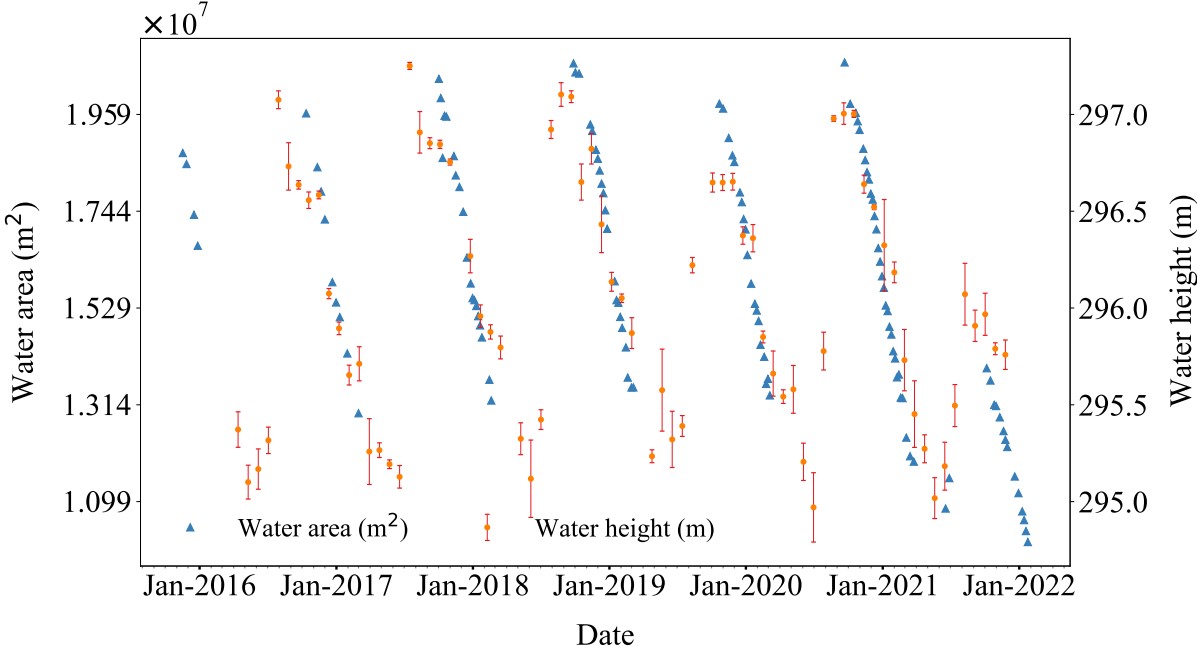

**Figure 2.** Time series of water areas (left y-axis) and water heights (right y-axis) with their associated median absolute deviation, for the Tibin reservoir (see Table 1), located in the center north of the Burkina Faso.

3) are considered outliers and are removed to obtain the final A-H curve, in line with Busker et al. (2019). The Root Mean
Squared Error (RMSE) and the R-squared ($R^2$) values are calculated to evaluate the accuracy of the regression.

Water heights are then estimated from water areas via the A-H curve, within the limits of the polynomial regression, i.e. without extrapolating the A-H curve. A filter is further applied to remove data with inconsistent variation (significant and rapid variations in the dry season for example) caused mainly by occasional mismatch between areas and heights. The final water height time series (Fig. 4) is composed of a combination of water heights directly obtained from the Sentinel-3 altimeter. It
includes both data used to estimate the A-H curve and data with no corresponding water area data, so not employed in the A-H curve, and water heights estimated through the A-H curve from water areas (from Sentinel-2).

### 3.5    Evaporation estimation

Gal et al. (2016) estimated the evaporation of a shallow lake (Agoufou) in Mali with the Penman equation (Penman and Keen, 1948). The context of this study being similar in climate, environment, and type of lakes, the same approach is used to estimate
evaporation using the Penman–Monteith equations and the methods by McMahon et al. (2013) (Suppl. S11). It requires the following meteorological data from the ERA5 dataset: downward surface solar radiation (J.m$^{-2}$), downward surface thermal radiation (J.m$^{-2}$), daily air temperature (K), daily dew point temperature (K), 10 m u-component and 10 m v-component of wind (m.s$^{-1}$) and altitude of wind speed measurement (m). It also requires altitude (m) derived from the Shuttle Radar

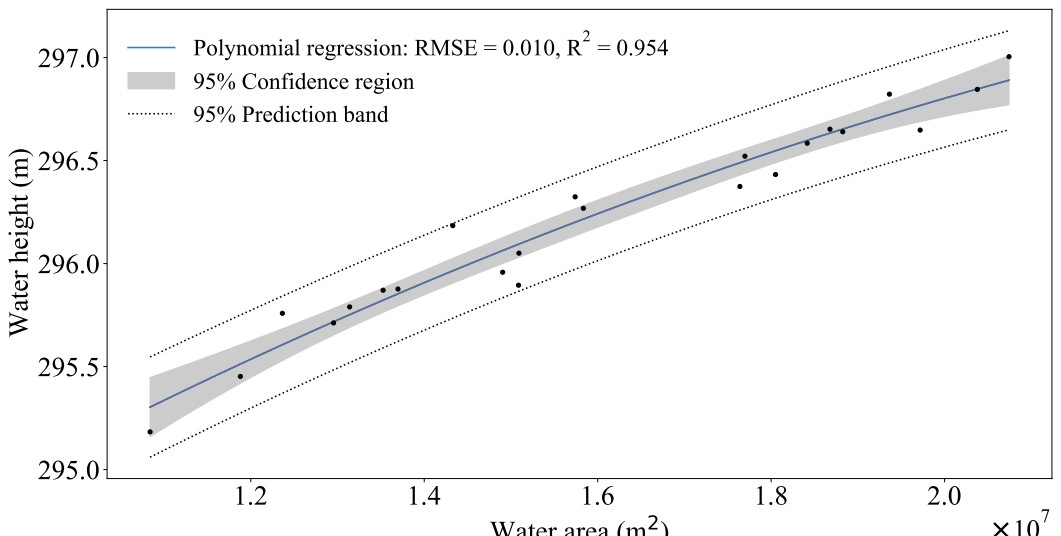

**Figure 3.** Water area-height curve (A-H) for the Tibin reservoir.

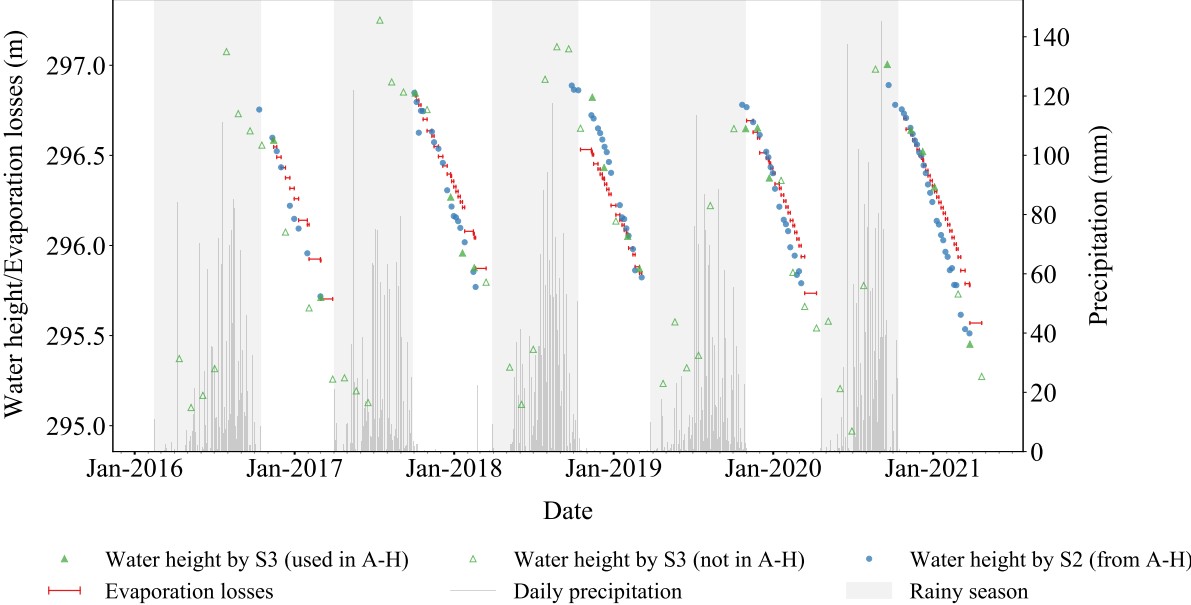

**Figure 4.** Water height time series from Sentinel-3 (S3) and Sentinel-2 (S2) data, along with the cumulated evaporation losses, daily rainfall and rainy season for Lake Tibin. The starting point to calculate evaporation losses is fixed to the water height at the first date in the dry season. The length of the red lines in the x-axis corresponds to the time between two successive water height samples.



Topography Mission (SRTM) DEM and the average albedo of the lake surface ($\alpha$), which is derived following Naegeli (2017)

as:

$$\alpha = 0.356\rho_{\text{red}} + 0.130\rho_{\text{NIR}} + 0.373\rho_{\text{SWIR1}} + 0.072\rho_{\text{SWIR2}} \tag{3}$$

from Sentinel-2 water reflectance ($\rho$). The start of the rainy season is taken as the day with rainfall exceeding 5 mm with rainfall in the following thirty days as well. The end of the rainy season is taken as the first day with rainfall below 5 mm also followed by sixty consecutive dry days.

## 170   4   Results

Over the whole study period, including the dry and the rainy season, the average water areas of the 42 lakes vary from 0.02 km$^2$ (Koankin) to 37.91 km$^2$ (Hagoundou) and averages to 5.28 km$^2$. We have identified that 69 % of the lakes turned out to be temporary lakes. Height seasonal variations vary from 4.86 m for the Boura reservoir in Burkina Faso in 2018, to 0.28 m for Lake Hagoundou in Mali in 2017, and averages to 1.94 m. Evaporation shows spatial variability which follows a latitude

gradient, with higher values in the North (with –7.04 mm.d$^{-1}$) than in the South (with –4.12 mm.d$^{-1}$) and averages –5.66 mm.d$^{-1}$ over the study period. The average albedo observed is 0.14 with a minimum value of 0.09 and a maximum value of 0.22.

### 4.1   Five-year averaged residuals water balance during the dry season

Of the 42 lakes studied, 37 have complete time series between 2016 and 2021 for which a five-year averaged residual water

balance, i.e. the difference between dry season water height changes rate and evaporation (Eq. 1), is estimated (Fig. 5 and Table 1). The five-year averaged residual water balance shows contrasted situations with values ranging from gains of 9.71 mm.d$^{-1}$ to losses of –12.45 mm.d$^{-1}$. 24 lakes, of which 75.0 % are located in Burkina Faso, have a residual water balance below –1 mm.d$^{-1}$. The Central Burkina Faso (red zone in Fig. 1) contains only lakes with a negative residual water balance of which 88.9 % have a highly negative residual water balance below –3 mm.d$^{-1}$. In northern Burkina Faso and near the western Niger

border (green zone in Fig. 1), 87.5 % of the lakes have a weak negative residual water balance, such as the Tibin reservoir illustrated previously (Fig. 4). Five lakes, all located in the Inner Niger Delta (blue zone in Fig. 1), have a positive residual water balance greater than 1 mm.d$^{-1}$. Finally, eight lakes display a residual water balance close to zero. They are located near the Niger River in the Tillabéry region of Niger, in southern Mali on the border with Niger and in the eastern part of the study area (orange zone in Fig. 1).


An example of water losses is given by a reservoir located in southern Burkina Faso (Manga in Table 1), close to the large lake of Bagré. The 2017–2018 dry season (Fig. 6) starts in late October 2017, when the lake water height is at 264.19 m and ends in February, when the water height is 262.60 m, i.e. a variation of 1.58 m. The evaporation losses are about twice as small as the height decrease, meaning that a significant part of the water losses is not due to evaporation. The residual water balance

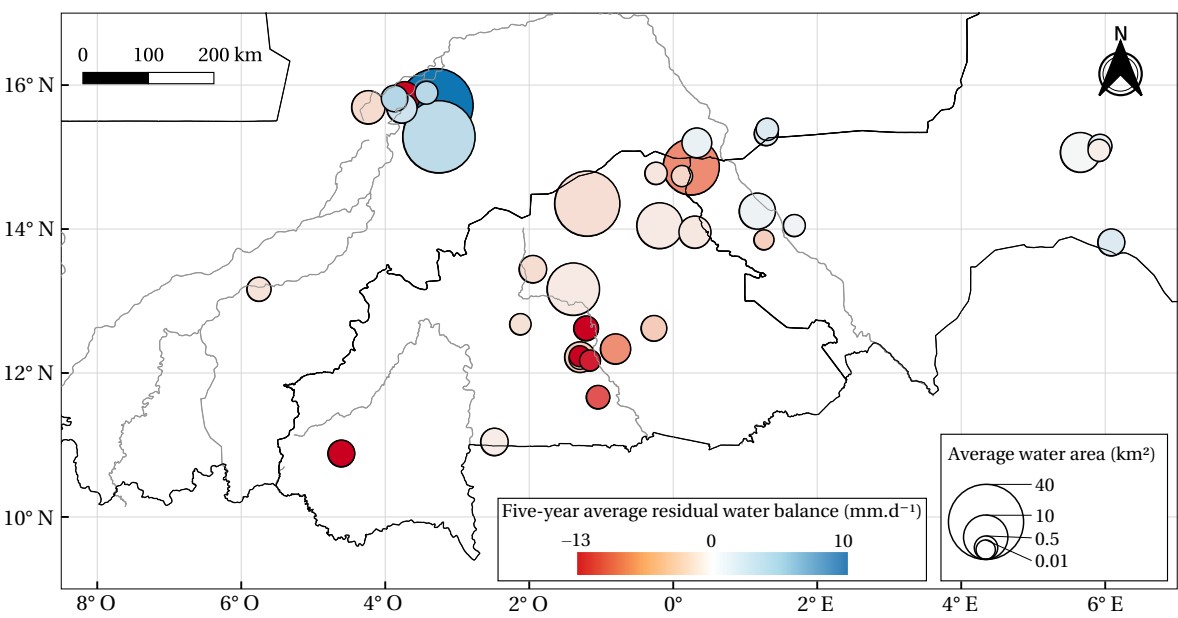

**Figure 5.** Five-year average residual water balance of each lake studied in Central Sahel from 2016 to 2021.

for 2017–2018 is –8.54 mm.d$^{-1}$ and averages –8.28 mm.d$^{-1}$ over the five years. False color images, during the dry season, show that the lake is surrounded by irrigated fields, which occupy an area similar to the lake's area. This suggests that most water losses are due to irrigation in this case (Fig. 7a and b).

     The water supply behaviour (positive residual water balance) is illustrated by a lake located in the Inner Niger Delta, where the Niger river splits in multiple reaches (lake Bakafé in Table 1, Fig. 8a, b and c). The 2019–2020 dry season illustrates this

case well (Fig. 9). Between October and May, water heights change from 262.14 m to 260.92 m, resulting in a 1.22 m decrease. However, the maximum height is reached about two months after the start of the dry season, indicating that precipitation is not the main cause of lake filling. Visual analysis of the Sentinel-2 images shows a connection between the lake and the river network, which is flooded from late October onwards (as in Fig. 8c). Once the peak of water height is reached, the lake empties approximately at the same rate as the estimated evaporation losses, since the two curves are parallel. The residual water balance

for 2019–2020 is 2.03 mm.d$^{-1}$ and averages 1.63 mm.d$^{-1}$ over the five years, which means that there is a regular dry season water inflow.

## 4.2    Interannual variability

The residual water balance can vary from year to year (Fig. 11) as a result of variability in anthropogenic management of resources, rainfall, length of the dry season, changes in inflow or outflow etc. Overall, the lakes do not show any trend in

residual water balance throughout the study period. About half of the reservoirs in Central Burkina Faso show greater losses in





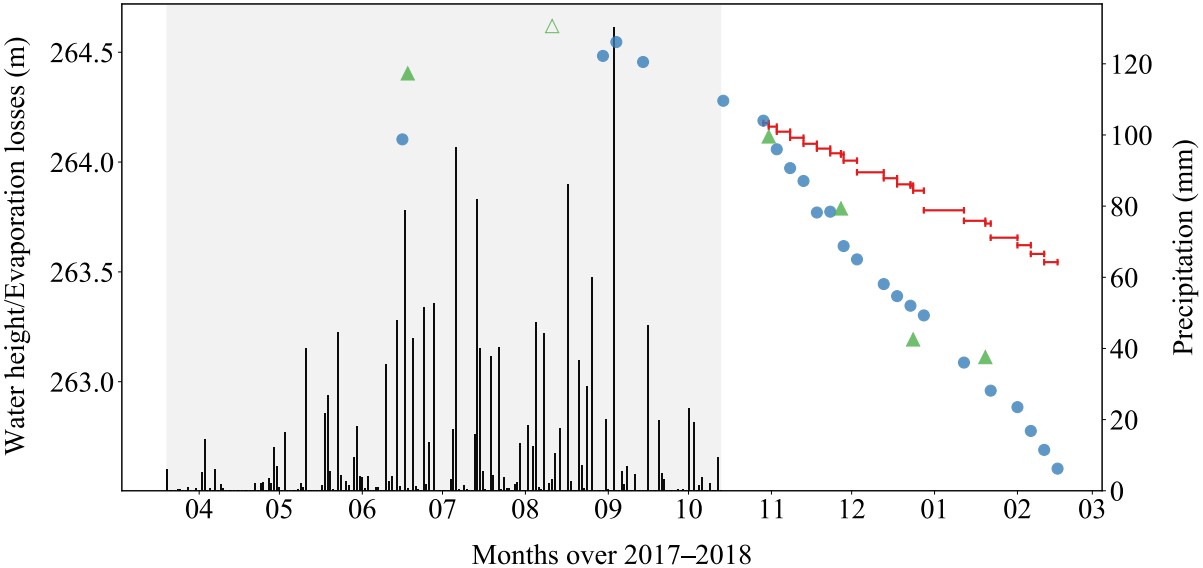

**Figure 6.** Water height time series for a lake with water loss behaviour (Manga), along with the cumulated evaporation losses, daily rainfall and rainy season.

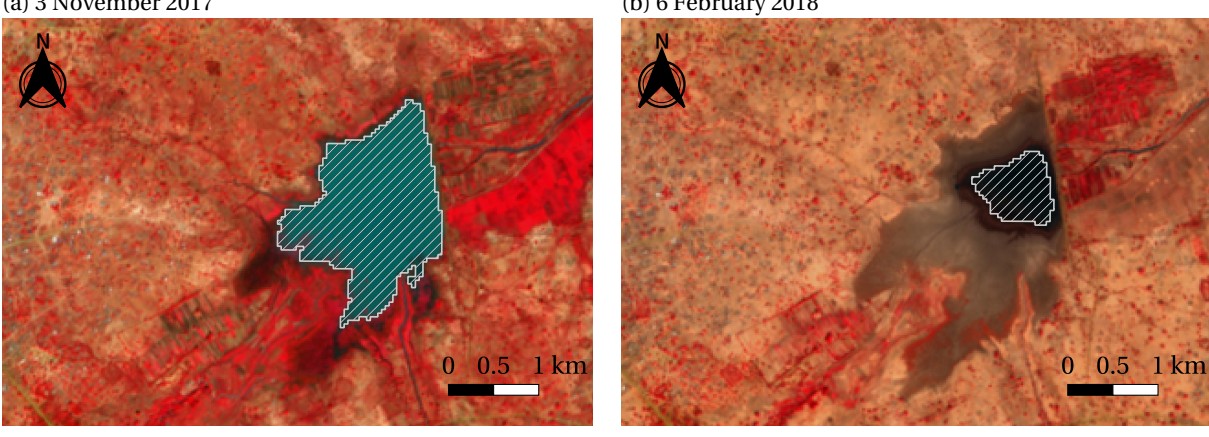

**Figure 7.** Sentinel-2 False Color images (NIR/Red/Green) of Manga reservoir surroundings with the lake contours obtained by thresholding on the MNDWI at (a) 27 November 2020 and (b) 21 January 2021. Wet vegetation is represented in red.



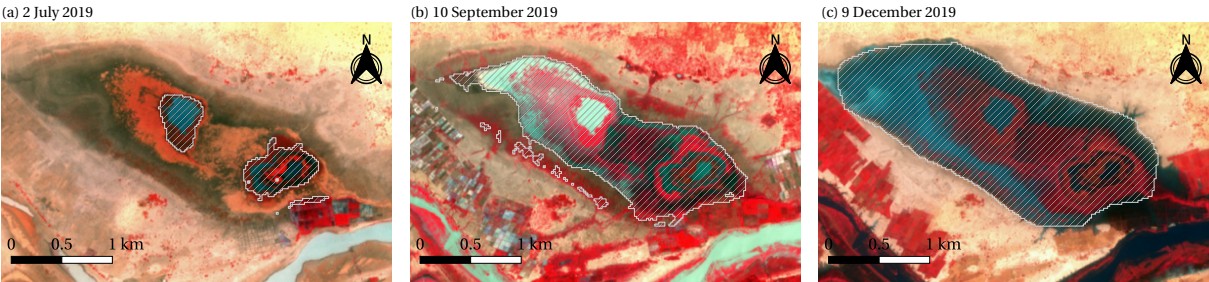

**Figure 8.** Sentinel-2 False Color images (NIR/Red/Green) of lake Bakafé surroundings with the lake contours obtained by thresholding on the MNDWI at (a) 2 July 2019, (b) 10 September 2019 and (c) 4 November 2019

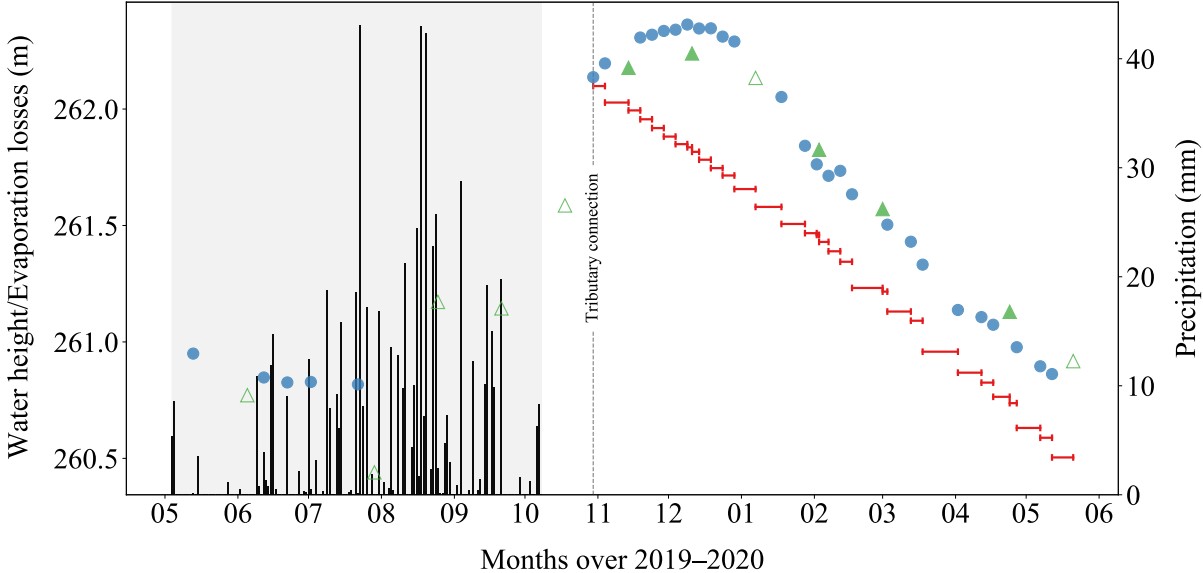

**Figure 9.** Same as Fig. 6 but for a lake with a water supply behaviour (Bakafé)

2020–2021 than the other years. The standard deviation (std) over five years has a minimum value of 0.27 mm.d$^{-1}$ (Lake N4) and a maximum value of 5.00 mm.d$^{-1}$ (Lake Yumban) and is equal to 1.86 mm.d$^{-1}$ on average over all lakes. Twenty-nine water bodies have a std greater than 1 mm.d$^{-1}$ and six lakes show a regime change switching between positive and negative values. Evaporation rate is quite constant over the five years for all lakes and its maximum standard deviation is equal to 0.42 mm.d$^{-1}$. Interannual water balance variability is sometimes caused by changes in dam functioning, like for the Gomde, a reservoir located in northern Burkina Faso. This reservoir was built to supply water needed by a gold mine, which is located to the southeast of the reservoir (Fig. 12), similar to the Tibin reservoir, created in 2012 for the Bissa gold mine (Newall, 2012; Ba, 2012). The standard deviation of the residual water balance over the five years is equal to 2.10 mm.d$^{-1}$ but the first two years show a residual water balance close to zero, while the last three years show important losses, with an average of –3.48





mm.d$^{-1}$, a drastic change which is seen also on the water height time series (Fig. 12). The maximum water height variation is obtained in 2020–2021 with 2.49 m and the minimum is in 2016–2017 with 0.84 m.

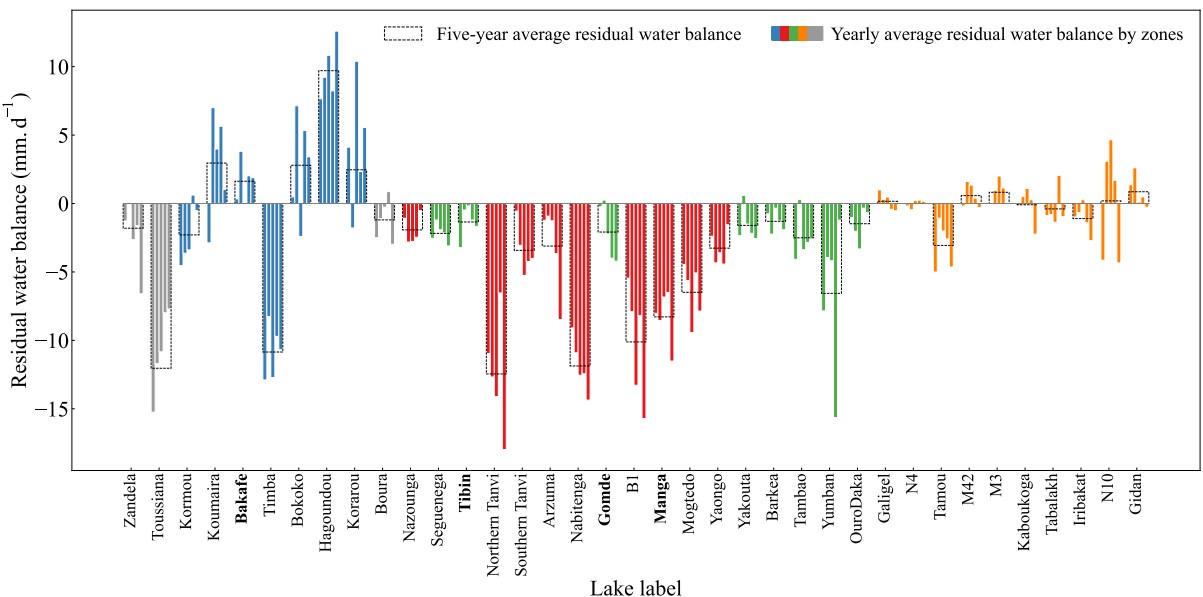

**Figure 10.** Yearly average of lake residual water balance from 2016 to 2021. Colored zones are defined in Fig. 1 and unclassified lakes are represented in grey. The labels on the x-axis in bold correspond to the lakes illustrated in this study.

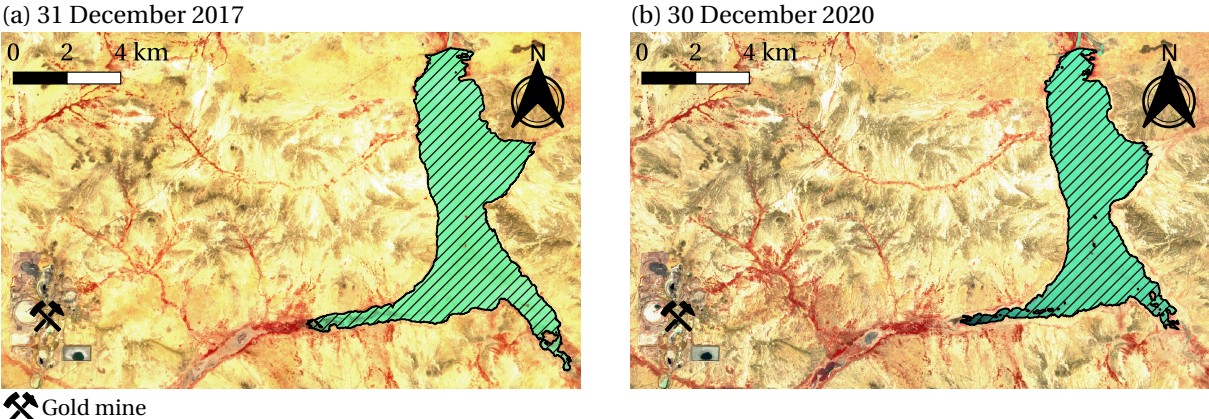

**Figure 11.** Sentinel-2 False Color images (NIR/Red/Green) of Gomde reservoir surroundings with the lake contours obtained by thresholding on the MNDWI at (a) 31 December 2017 and (b) 30 December 2020.



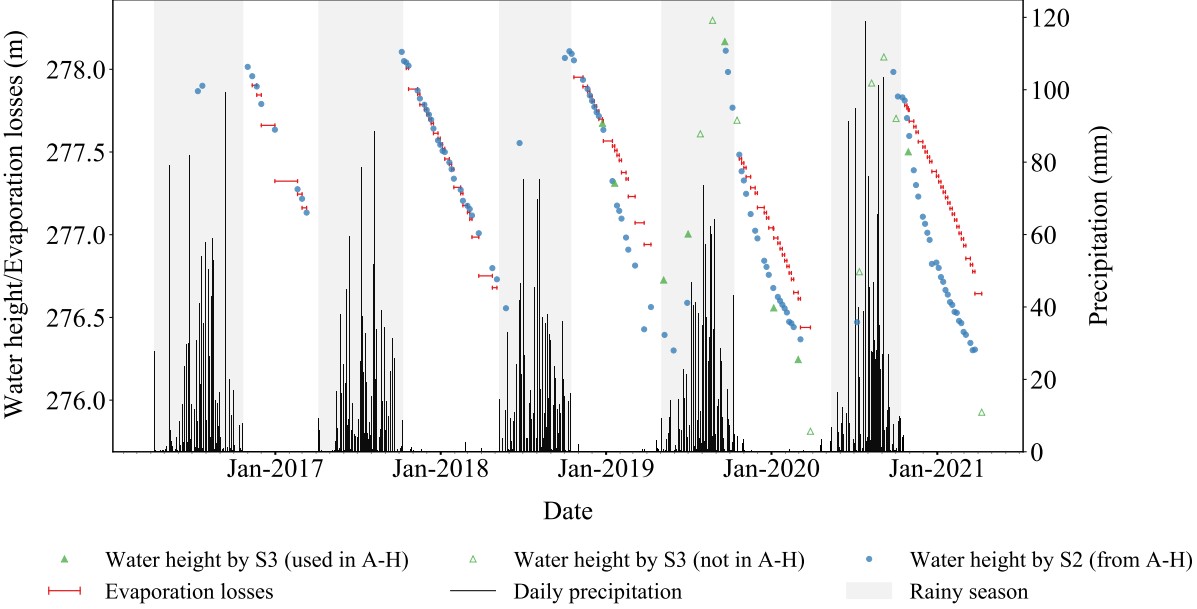

**Figure 12.** Water height time series from Sentinel-3 (S3) and Sentinel-2 (S2) data, along with the cumulative evaporation losses, daily rainfall, and rainy season for Lake Gomde. The starting point to calculate evaporation losses is fixed to the water height at the first date in the dry season. The length of the red lines in the x-axis corresponds to the time between two successive water height samples.

## 5 Discussion

The combination of altimetry, time series of optical images, and evaporation modelling reveals a large variety of situations and different hydrological regimes. The studied lakes are located in diverse environments. For instance, watersheds are dominated
by rainfed or irrigated crops or natural vegetation, growing on sandy or rocky soils. Lakes also differ in terms of their characteristics, such as whether they are open water or harboring dense aquatic vegetation, if there are trees growing in the flooded areas, or if the water is clear or extremely turbid, which is associated with very high water albedo. This complexity results in uncertainties in water regime calculation (Eq. 1). First, aquatic vegetation leads to radiometric variability of water pixels which makes it difficult to use automatic algorithms for water detection. Moreover, lakes drying up may also affect classification.
These water detection difficulties were recently pointed out by Reis et al. (2021) and in the same line, Ogilvie et al. (2018) showed that in central Tunisia, the Global Surface Water dataset (Pekel et al., 2016) had an omission error rate of 41 % on shallow lakes, mostly due to pixels with vegetation or algae. Moreover, the albedo values (Table 1) highlight the diversity of water colour with values ranging from 0.09 for dark clear surface water, to 0.22 for bright turbid surface water, which is higher than the albedo values generally found in lake studies. McMahon et al. (2013) for instance suggest a default value of albedo of
0.08 to compute the evaporation of open water.



This variability challenges the determination of an optimum MNDWI threshold. In this work, the threshold is chosen ad hoc for each lake (Sect. 3.3) as recommended by Reis et al. (2021) and a constant threshold throughout the study period proved to be efficient for our study. However, the method developed here is not very sensitive to systematic errors in water body surface
area detection. Since areas are used to interpolate water heights, via hypsometric relationships, there is no need to derive an estimation of water area for the whole lake and part of lake can be discarded, provided this is systematic. This situation is encountered for some lakes without a well identified connection to a river or with another lake or lakes that overflow, for ex-ample downstream of a dam, which can be truncated to avoid misclassifications. Similarly, as long as the threshold applied to the MNDWI consistently detects water pixels even in the presence of vegetation, the time series of water areas has consistent
variations even if the absolute value of the water area may not be correct. Despite of this, water area classification remains a source of error in the calculation of the residual water balance, occasionally creating outliers in the water area time series. A close inspection of these cases points to misclassification of aquatic vegetation for some images. As a result, 19 lakes under the altimeter tracks could not be included in the study because non-systematic detection problems in water areas made the time series too noisy.


In addition, several other lakes were discarded because of noisy or inaccurate water height time series. This is caused by the radar altimeter resolution along-track of 300 m that only allows detection of lakes larger or similar to its resolution. The retracking algorithm failures (the retracking being designed for ice surface, Crétaux et al., 2018) or the prolonged drying of certain lakes for entire years also do not allow to have enough data to construct the hypsometric curve (ephemeral lakes or
ponds). Moreover, other water-like sources in the altimeter footprint sometimes contaminate the signal (Jiang et al., 2020), which is possible due to the across-track resolution of 1.64 km and shifting of the track up to 1 km. The different filtering processes allow for the elimination of remaining outliers.

The seasonal amplitude, averaged over the period 2015–2021 for all lakes, ranges between 0.89 m and 3.23 m with a median
value of 2.06 m. This is higher than the value found by Cooley et al. (2021), who estimated a median of 1.60 m with a 1.49 m, 1.71 m range for 127 lakes in the Niger basin using two years of ICESat-2 data. The temporal resolution of ICESat-2 time series of about 91 days may miss the maximum and minimum heights of most lakes, even if the larger lakes may cross several ICESat-2 tracks, pointing to the importance of having finer temporal resolution. In addition, our study samples a larger variety of hydrological behaviours and analyses lakes that are not found in the global database employed by Cooley et al. (2021) and
other global studies.

Evaporation is an important term of the lake water balance. We have compared results obtained with the Penman–Monteith method with the evaporation derived from pan observations available for the Boura reservoir and with the GLEV method. Multi annual averages, from 2012 to 2014, were equal to 5.33 mm.d$^{-1}$ by our estimation, 5.40 mm.d$^{-1}$ by derived evaporation
(Fowe et al., 2015) and 5.38 mm.d$^{-1}$ by GLEV (Zhao et al., 2022). Evaporation differences by these three methods are lower



than an evaporation uncertainty of $\pm$ 1 mm.d$^{-1}$ considered by Gal et al. (2016) for the Penman method.

Finally, the calculation of the dry season annual residual water balance is impacted by the first and last data at the beginning and end of the dry season. A correct estimation of these values is therefore important, otherwise some fluxes may be over-

seen, like early dry season filling by rivers. Overall, we consider that the residual water balance variations above 1 mm.d$^{-1}$ are unlikely to be caused by errors, but rather indicates water inflows in the dry season, whereas a residual water balance below –1 mm.d$^{-1}$ would point towards water losses. The five-year averaged residual water balance shows consistent spatial patterns. In the Inner Niger Delta, waters bodies show a predominantly water supply behaviour. Indeed, in this area lakes are connected to the river network, and they are filled initially by rainfall and then to a greater extent by river waters coming from

the upper Niger watershed, causing dry season flooding of the delta (Olivry, 1995). Sometimes two processes successively dominate a lake water regime, such as a water supply at the beginning of the dry season and a water loss afterwards. Lakes in the eastern part of the study region (from about 0° E) show a weak positive residual water balance. Even if these low values are below the uncertainty that we estimate, a limited water supply during the dry season would be in line with water supplied by groundwater in this region (Favreau et al., 2009). In Burkina Faso, and more importantly in the centre, a water loss signal

prevails. These observations are in line with Fowe et al. (2015) and Venot and Krishnan (2011), who show that the variations of small reservoirs in this region are due to water withdrawn for small scale irrigation, which is usually detected by the growth of surrounding crops during the dry season. Exchanges with groundwater (Sophocleous, 2002) could also lead to losses due to infiltration through the lake bottom. This is more likely to occur at the beginning of the dry season, when lakes area is high and banks are flooded, whereas later on water losses are less significant because lake bottoms are usually silted. Further north

in Burkina Faso, near the border with Mali and Niger, water bodies show little or no residual water balance loss, which is consistent with limited anthropogenic actions over these reservoirs. The fact that several reservoirs in this area show residual water balance close to zero moderates the conclusions of Cooley et al. (2021) on the substantial influence exerted by humans on surface water storage variability. Our results show very different regimes within the same catchment, so it is complicated to apply variations at a catchment scale to lakes within that catchment. The reservoirs in our study area are sometimes reported to

have low performance (Venot and Cecchi, 2011), mainly due to limited funding and human resources put into reservoir management (Frenken, 2005). Some of these reservoirs were built for gold mining, so the absence of anthropogenic withdrawals may seem inconsistent. However, this area suffers from serious security issues. Since 2015, the number of armed conflicts has been increasing and is now spreading to the whole region. Discussions with colleagues in Burkina Faso (J.M. Dipama, pers. comm.) and search of the local press, lead to the hypothesis that part of the population living near these reservoirs has moved to

avoid conflicts, leaving reservoirs with little manpower and limited irrigation projects. Another example of the possible impact of conflicts in this area is the Gomde reservoir, which shows a significant change in the residual water balance after 2018 (close to zero before 2018 and significantly negative afterwards). In this case, attacks by armed groups interested in the gold mines (Assanvo et al., 2019) are the probable cause of damages to the dykes, leading to dam leakage since 2019.





This study illustrates the potential of recent remote sensing sensors to explore the hydrological behaviour of lakes in semi-arid areas. Although it is not yet possible to identify and quantify all fluxes, the residual water balance approach provides very valuable information on surface water resources at the regional scale. The spatio-temporal resolution of current satellites allows monitoring of small water bodies. The methodology developed here, based on freely available data and tools, is easily transposable to other regions with similar climate. Currently, the water balance estimation is restricted to lakes below the

altimeters tracks, which may be around 1 % or less of the total number of lakes in this region. For example, of 1650 reservoirs analysed by Cecchi et al. (2009) in Burkina Faso, only 21 are surveyed in this study. Only one among the lakes studied is found in the DAHITI database and none in the HYDROWEB and G-REALM databases, and all three are usually employed in global studies. A common approach to address water bodies for which water level estimations are not available is to apply the same hyspometric curve for all lakes in a given region (Cooley et al., 2021; Hou et al., 2022), assuming that in similar geological

situations their shapes are not very different. For the method developed here it is however essential to derive A-H curves for each lake since applying a general one for all water bodies in our study region would result in misleading quantification of water fluxes. With the arrival of the Surface Water and Ocean Topography (SWOT), which will be launched at the end of 2022, the number of lakes that can be monitored to assess water height changes will greatly increase (Grippa et al., 2019).

## 6   Conclusions

In this study, a method to estimate the hydrological regime of 37 small water bodies from 0.04 km$^2$ to 37.91 km$^2$ in Central Sahel was proposed based on remote sensing data from 2016 to 2021. The method combines Sentinel-3 and Sentinel-2 for the water height and water area respectively, with meteorological variables from ERA5, and ancillary data from multiple sensors. A dry season water balance is estimated over five years for each lake by comparing evaporation and water height changes, which characterizes lake hydrological regime. This method allows for a large-scale study of many ungauged water bodies,

including small ones and lakes with aquatic vegetation cover, that are frequently overlooked in large scale studies. Lakes showing dry season water losses (where water depletion is greater than evaporation) were mainly found in central Burkina Faso. This behaviour also concerns lakes in the north of the country, but to a lesser extent. In the Inner Niger Delta, lakes mostly show dry season water supply, caused by water inflow from multiple river networks during flooding of the delta, and filling the lakes generally at the start of the dry season. Other lakes display a balanced behaviour, where water height closely follows the

evaporation rate. The limited water supply observed for lakes in Niger may be caused by exchanges with groundwater, which has been observed in this region. Interannual variations of lake hydrological regimes have been observed, with some significant changes attributed to changes in the anthropogenic use of water resources.

*Data availability.*   The Sentinel-2 data MultiSpectral Instrument (MSI) are available on Google Earth Engine (GEE, Gorelick et al., 2017)

through the "Sentinel-2 MSI: MultiSpectral Instrument, Level-1C" collection (https://developers.google.com/earth-engine/datasets/catalog/



COPERNICUS_S2_HARMONIZED). The Sentinel-3 Sar Radar Altimeter (SRAL) data and the Altimetric Time Series Software (Al-TiS, Frappart et al., 2021) are obtained from the Centre de Topographie des Océans et de l'Hydrosphère (CTOH, http://ctoh.legos.obs-mip.fr/data/offline-request-form). Meteorological data are obtained from the European Centre for Medium-Range Weather Forecasts (ECMWF) of the Copernicus Climate Change Service (C3S) through the "ERA5 reanalysis hourly data on single levels from 1959 to present" database
(Hersbach et al., 2018). The results contain modified Copernicus Sentinel and C3S information (2022). Neither the European Commission nor ECMWF is responsible for any use that may be made of the Copernicus information or data it contains. Precipitations are available on GEE through the "GPM: Global Precipitation Measurement (GPM)v6" collection (Huffman et al., 2019, https://developers.google.com/earth-engine/datasets/catalog/NASA_GPM_L3_IMERG_V06). Evaporation rate data are retrieved from the Global Lake Evaporation Volume dataset (GLEV, Zhao et al., 2022, https://doi.org/10.5281/zenodo.4646621) and from Fowe et al. (2015).

*Author contributions.* All authors contributed to the writing and design of the study.

*Competing interests.* The authors declare that they have no conflict of interest.

*Acknowledgements.* We acknowledge Fréderic Frappart and Fabien Blarel for their involvement in the processing of data through the AlTiS software, Jean-François Crétaux for advice on the area-height curve derivation, and Félix Girard for discussion of our water detection and altimetry methods. We thank Hedwige Nikiema and Jean-Marie Dipama for their information on the region and lakes, and Yasmin Fitts for
her valuable proof-reading of the manuscript.

*Financial support.* This work was supported by CNES, focused on the prelaunch activities in the framework of SWOT mission.





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
