# Peer review of "Hydrological regime of Sahelian small water bodies from combined Sentinel-2 MSI and Sentinel-3 SRAL data"

_Hydrology and Earth System Sciences, 2022_

## Author Response (AR1)

**Authors' Response to Reviews of**

**Hydrological regime of Sahelian small water bodies from combined Sentinel-2 MSI and Sentinel-3 SRAL data**

Mathilde de Fleury, Laurent Kergoat, and Manuela Grippa
*Hydrol. Earth Syst. Sci. Discuss., https://doi.org/10.5194/hess-2022-367, 2022,*
* * *
**RC:** *Reviewers' Comment*,     AR: Authors' Response,     □ Manuscript Text

We thank the referees for their careful reviews and very valuable comments and suggestions. Following referee's suggestion, we have been able to obtain ground validation data. These data are now included as supplementary information in the revised manuscript. Here is a detailed response to comments, along with the corresponding changes in the manuscript (we also made a few minor modifications in the acknowledgements, references, etc).

**1.  Referee #1**

**RC:** *General comments:*
*I regret that the authors did not validate some of their results, such as the relationships between water heights and water areas of reservoirs with ground truth monitoring data (field measurements) or initial A-H curves.*

AR:  We have been able to obtain existing ground truth for two studied lakes in Burkina Faso from the Direction Générale des Ressources en Eau (DGRE) via l'Institut International de l'Eau et de l'Environnement (2IE). We also used a published curve for Lake Bam (Pouyaud, 1975). We have added in the Supplement the comparison of in situ and satellite derived V-H curves and an estimation of errors on retrieved H. We agree it really add values to our work, thank you for the suggestion.

**RC:** *Page 5, line 118: Equation (1)*
*There is a problem in equation (1) regarding the sign of evaporation in the residual water balance. Precipitation and evaporation fluxes are considered as an inflow and outflow respectively, therefore cannot have the same sign in equation (1). Please check this. If so, I hope the results are not affected.*

AR:  Indeed, in our calculations we considered evaporation as a loss but the equation in the manuscript was not well written. Thank you for having pointed out this error. The equation has been corrected as follows:

$$R = \frac{1}{t_1 - t_0} \left[ \Delta H_{t_0, t_1} - \sum_{i=t_0}^{t_1} (P_i - E_i) \right] \tag{1}$$

**RC:** *Page 5, line 123*
*The authors mentioned that the relationship between the water heights and water areas of a reservoir is also called "hypsometric curve". I am not sure that this relationship is interpreted in the same way as the hypsometric curve, please check.*

AR:  To avoid confusion, we have removed the term hypsometric curve (page 5, line 123; page 16, line 240; page 16, line 254; page 18, line 313).

**RC:** *Page 6, Table 1*
*In Table 1, it would be interesting to associate the uncertainties on the average values of the different variables*

**AR:** In the discussion we mention the difficulties in establishing the uncertainties for each variable and each lake. We have listed the sources of these uncertainties but it is not straightforward to quantify them for each lake without dedicated field survey. However, as suggested, we performed an estimation of the errors on the final water height time series (see Supplement) which include errors on both water height and area estimation. We have found quite low values of RMSE equal to 0.073 m for Lake Seguenega and 0.015 m for Lake Bam, which give confidence on our results.

**RC:** *Page 8, lines 153-156*
*Please, rewrite both sentences to clarify the final water height time series.*

**AR:** This has been done. Original sentences: " The final water height time series (Fig. 4) is composed of a combination of water heights directly obtained from the Sentinel-3 altimeter. It includes both data used to estimate the A-H curve and data with no corresponding water area data, so not employed in the A-H curve, and water heights estimated through the A-H curve from water areas (from Sentinel-2). "
Modifications: The final water height time series (Fig. 4) is composed of water heights derived from Sentinel-3 and water heights estimated from Sentinel-2, through the A-H curve. All altimetry derived water heights are considered, even those not used to build the A-H curve.

**RC:** *Page 10, line 175*
*Please, can you explain the negative values of daily evaporation from the reservoirs?*

**AR:** These were evaporation 'losses', but to avoid confusion we have replaced the negative values with positive values of evaporation.

**RC:** *Page 10, lines 179-180*
*Please, check the definition of the residual water balance of a reservoir.*

**AR:** To avoid confusion we deleted the definition here and left the explanation of the residual water balance provided in Section 3.1. So the modified sentence reads as: "five-year averaged residual water balance  (Eq. 1)"

**RC:** *Page 12, Figure 6*
*Please add a legend. For the X-axis labels, it would be good to write the months in letters followed by the year*

**AR:** The legend and axis were modified according to your comment.

**RC:** *Page 12, Figure 7b*
*In Figure 7b, please check the date of the image. Is it 6 February 2018 or 21 January 2021?*

**AR:** This was a mistake, sorry about it. The errors in the figure caption were fixed.

**RC:** *Page 13, Figure 8c*
*In Figure 8c, please check the date of the image. Is it 9 December 2019 or 4 November 2019?*

**AR:** The errors in the figure caption were fixed.

**RC:** *Page 13, Figure 9*
*Please add a legend. For the X-axis labels, it would be good to write the months in letters followed by the*

*year*

AR: The legend has been added and x-axis labels were modified according to your comment.

**RC:** *Page 15, lines 224-225*
*This sentence needs to be clarified. Are these the watersheds of the lakes studied?*

AR: This has been done. Original in manuscript: "The studied lakes are located in diverse environments. For instance, watersheds are dominated they can be surrounded by rainfed or irrigated crops or natural vegetation, growing on sandy or rocky soils."
The text was modified as follows (modifications in red): "The studied lakes are located in diverse environments. For instance,  they can be surrounded by bare or vegetated areas (rainfed crops, irrigated crops, natural vegetation) and by soils with different hydraulic characteristics (sandy, loamy, rocky soils).

**RC:** *Page 16, lines 239-240*
*Please, the authors should explain why the method developed is not sensitive to systematic errors in water body surface detection. Can the authors give some examples of errors? For example, the presence of waves in the reservoir is not discussed at all. Even if waves are not significant at the study area, it should be explicitly commented on given that the methodology is based significantly on the determination of water heights and water areas of reservoirs.*

AR: Large random errors (other than systematic errors), will be easily identifiable as outliers in the time series and will not remain after filtering. These errors may be due to waves indeed, currents, winds or antropogenic activities (Frappart et al., 2021). Systematic errors in surface water area are commented below.

**RC:** *Page 16, lines 240-241*
*The authors mentioned that "There is no need to derive an estimation of water area of the whole lake". What would be the impact of underestimating the surface areas of water bodies on water heights? Could it be significant or not?*

AR: We agree that this point needs clarification, since it is not straightforward. We therefore modified the text as follows (modifications in red): "However, the method developed here is not very sensitive to systematic errors in water body surface area detection.  Given that water areas are only used to estimate water heights via the A-H curve, systematic errors in water area detection will not affect the final height estimation. For example, systematically missing a part of the lake in the water area detection (truncation) will modify the absolute water area values in the A-H curve but will not change the water height values. This situation is encountered for some lakes without a well identified connection to a river or with another lake or lakes that overflow, for example downstream of a dam. "

Figure AR1, below, shows a case study that illustrates our point.

[Figure]

Figure AR1: Case study of the impact of lake area underestimation. Figure a) represents the lake with a smaller polygon (truncation), over which water areas are purposely underestimated (pink area). Figure b) represents the water surfaces estimated using MNDWI thresholding over the full lake and over the restricted pink polygon. Figure c) represents the respective A-H curves. Figure d) represents the water heights reconstructed time series. The results with the entire lake (blue and called Tibin), are similar to the result with the truncated lake (orange and called Tibin truncated). This is because the method uses changes in surface area to interpolated water height.

**RC:** *Page 16, lines 260-261*
*Please, rewrite the sentence*

AR: Original sentence in manuscript: "This is higher than the value found by Cooley et al. (2021), who estimated a median of 1.60 m with a 1.49 m, 1.71 m range for 127 lakes in the Niger basin using two years of ICESat-2 data."
Revised manuscript (modifications in red): "This is higher than the values reported by Cooley et al. (2021), who estimated water height amplitudes over 127 lakes in the Niger basin ranging between 1.49 m and 1.71 m (median 1.60 m) using two years of ICESat-2 data."

**RC:** *There is a couple of references that are wrong!! (i.e. missing in the text of the manuscript, although listed in the list of references).*

AR: Thank you for pointing this. The list of references was fixed. We deleted unused references: Bader et al., 2011; Gao 2015. We modified the citations of references: Abdourhamane Touré et al., 2016; Douxchamps et al., 2014. We corrected the dates in the reference: Huffman et al., 2019.

**2. Referee #2**

*Introduction*

**RC:** *Line 28 The authors write an interesting comment about the lack of information about temporary water bodies, but there is no defined (or written) methodology for segmenting temporary water bodies*

**AR:** We added this in Section 3.3 as suggested (modifications in red): "The MNDWI threshold (Table 1) is chosen ad hoc for each lake and kept constant over the study period. This method ensures that water pixels are not detected when water bodies dry up."

**RC:** *Line 39: Previous studies cite an RMSE of 0.67 m, which is quite large for water level accuracy, as systems are expected to have much higher accuracies, closer to 0.10 m. This study does not validate the water level measurements. The RMSE that is reported (Figure 3) is the RMSE of the model fit to the data (assuming the altimetry data is correct). This potential 0.67m difference is important when comparing results with other studies.*

**AR:** As suggested by both referees, we evaluated water heights of our time series using in situ data available for two studied lakes. This evaluation has been included in the Supplement in the revised manuscript. We found RMSE of 0.073 m and 0.015 m on estimated water heights for these two lakes. This is indeed closer to 0.10 m and smaller than the 0.67 m found by Normandin et al. (2018, Suppl. Table S5).
We have added this information in Section 3.4: "We have carried out validations of our A-H curves with in situ data (Supplement) over two lakes (Seguenega and Bam, Table 1). We obtained RMSE of 0.073 m and 0.015 m, and biases of –0.070 and 0.006."
We also added a sentence in Discussion: "All the different filtering processes allowed to reduce the errors on the final water height time series. The RMSE value obtained by comparing to in situ data are considerably lower than the value 0.67 m reported by Normandin et al. (2018, Suppl. Table S5)."

**RC:** *Line 42: It is not recommended to cite Cooley et al 2021 for ICESat-2 or G-REALM as this group is not responsible for producing these data.*
*Citations for G-REALM are suggested to include Birkett et al 2010 and Birkett et al 2017, and the USDA site directly [`https://ipad.fas.usda.gov/cropexplorer/global_reservoir/Default.aspx`]. The sentence about laser altimetry is suggested to be re-written as 'Studies have demonstrated using laser altimetry from ICESat-2 to measure water levels (Cooley et al 2021, (and others)).' This suggestion is to clarify the relevance of that study to the present study, and not to confuse the production of the ICESat-2 data with the utilization of the data for the water level mapping study.*

**AR:** Thank you for this clarification. The references and sentences were modified according to the referee comments. Original sentence: "Laser altimetry with ICESat-2 (Cooley et al., 2021) is also a technique currently used to measure water levels."
Modifications (in red): "Laser altimetry data from ICESat-2 have been used to derive water level changes (Cooley et al., 2021)."

*Section 3.2 Lake Water Height Estimation*

**RC:** *Please specify the frequency and sensor (Sentinel-3 Ku/C-band) used in AlTiS, to provide a reference for the backscatter coefficient.*

AR: This has been done. Modifications in red: "Water heights are obtained from Sentinel-3 SRAL (Ku-band at 20 Hz) data"

***Section 3.3 Surface Water Area Estimation***

**RC:** ***It makes sense that the water index thresholds might vary by water body. Additionally, it would be a useful contribution to understanding the hydrological regimes if the mNDWI thresholds could be used to explain the variability between groupings of similarly reflecting lakes. (For example, grouping lakes where mNDWI==0, mNDWI> 0.1, mNDWI< -0.1)***

AR: This is an interesting idea. We have tested it but we did not find any correlation between the MNDWI threshold and the value of the five-year average residual water balance (Fig. AR2 below). Moreover, looking at the spatial distribution of the MNDWI threshold (Fig. AR3 below), we also cannot conclude about patterns reflecting the hydrological regime or the typology of water bodies (e.g. connection to rivers, large lakes, small lake, turbid, clear, ect) and the threshold retained. It should be noted that for a given lake, there is usually a range of MNDWI thresholds that work equally well. This may explain why we do not find evident relationship between MNDWI threshold and water bodies characteristics.

[Figure]

Figure AR2: Five-year average residual water balance compared to MNDWI threshold for each lake

[Figure]

Figure AR3: Map of the MNDWI threshold

*Figure 2*

**RC:** *Please provide a description of the symbols in the figure caption.*

AR: This has been done, modifications in red: "Time series of water areas (left y-axis, in blue) and water heights (right y-axis, in orange) with their associated median absolute deviation (orange bars), for the Tibin reservoir (see Table 1), located in the center north of the Burkina Faso."

*Section 3.5 Evaporation estimation*

**RC:** *The Penman-Monteith equation does not require meteorological or DEM data from any specific source, such as SRTM or ERA. It is suggested to edit the sentence to say that the "Penman-Monteith equations... require the following meteorological and elevation data variables such as [var0, var1, var2, var3... ]. Here, the meteorological variables are identified from ERA5, and the DEM is from SRTM."*

AR: We added the following modifications in red according to the referee suggestion: "It requires the following meteorological data, which we extracted from the ERA5 archive".

**RC:** *The last two sentences mention the rainy season timing; it is not clear how that relates to the discussion of the input variables since the seasonal timing was not mentioned previously in the context of the P-M equations.*

AR: Our study is focused on the dry season, so the two sentences has been changed accordingly (modifications in red): "To calculate evaporation over the dry season period only, we estimate start and end dates from

rainfall data. The end of the dry season is taken as the date of the first day with rainfall exceeding 5 mm and followed by at least another  rainfall exceeding 5 mm in the following thirty days . The start of the dry season is taken as the first day with rainfall below 5 mm  followed by sixty consecutive dry days."

*Section 4*

**RC:** *That 69% of the lakes turned out to be temporary lakes is interesting. The authors mentioned the importance of studying temporary lakes at the beginning of the article, but there was no mention in the lake identification section or in the methods that described how these temporary lakes would be identified or assessed. The majority of the lakes are temporary, it would be useful to have more of a description of them, perhaps in Section 3.3., where the spectral properties of the lakes are being assessed.*

AR: The adaptive thresholding method ensures that no water is detected when the lake is dry. Moreover, we exclude altimetry data for which the corresponding lake water area is equal to zero. We have added the following modifications (in red), in Section 3.3 to better explain this: "The MNDWI threshold (Table 1) is chosen ad hoc for each lake and kept constant over the study period. This method ensures that water pixels are not detected when water bodies dry up."

*Section 4.1 Five-year averaged residual water balance*

**RC:** *Is it possible to add the colored boundaries from figure 1 to figure 5? This would make it easier to reference rather than going back up to the previous figure.*

AR: This has been done. We also updated the links to Fig. 5.

*Figure 5*

**RC:** *Please provide a description of the symbols in the figure caption.*

AR: We fixed the caption as suggested, the modifications are in red: "Five-year average residual water balance of each lake studied in Central Sahel from 2016 to 2021. The circles representing the studied lakes have an area proportional to the lake average water area. Lakes with water supply and water loss behaviours appear in blue and red respectively."

*Figure 6*

**RC:** *Please provide a legend in the figure as well as a description of the symbols in the figure caption.*

AR: We fixed the legend and caption as suggested, the modifications are in red: "Water height time series (in blue and green) for a lake with water loss behaviour (Manga), along with the cumulated evaporation losses (in red), daily rainfall (in black) and rainy season (in gray)."

*Figure 7*

**RC:** *Please provide a description of the shaded region in the caption (assuming it is the identified open water extent).*

**AR:** Caption with modifications in red: "Sentinel-2 False Color images (NIR/Red/Green) of Manga reservoir surroundings with the lake contours (in white) obtained by thresholding on the MNDWI at (a) 27 November 2020 and (b) 21 January 2021. Active vegetation appears in red."

**RC:** *Figure 8 and Figure 9, the same comments as Figures 7 and 6 (requesting more labels).*

**AR:** Caption with modifications in red: "Sentinel-2 False Color images (NIR/Red/Green) of lake Bakafé surroundings with the lake contours (in white) obtained by thresholding on the MNDWI at (a) 2 July 2019, (b) 10 September 2019 and (c) 4 November 2019".
We changed the legend and caption as suggested, the caption now reads as follow: "Water height time series (in blue and green) for a lake with water supply behaviour (Bakafé), along with the cumulated evaporation losses (in red), daily rainfall (in black) and rainy season (in gray)".

*Section 5 Discussion*

**RC:** *Lines 225-228: "Lakes also differ in terms of their characteristics. . . This complexity results in uncertainties in water regime calculation (Eq. 1)." I suspect that this comment might better relate to the mNDWI calculation where the water bodies are identified or the albedo calculation. It is true that the water surface variability or uncertainty will lead to uncertainty in the water balance, but it is important to point out that the source of the uncertainty is the mNDWI water identification (Eq 2), not in Eq 1 itself.*

**AR:** These sentences were modified to take into accounts suggestions (in red): "Lakes also differ in terms of their characteristics, such as whether they are open water or harboring dense aquatic vegetation, if there are trees growing in the flooded areas, or if the water is clear or extremely turbid, which is associated with very high water albedo. This complexity results in uncertainties in water detection by MNDWI thresholding calculation (Eq. 2) which impact water regime calculation (Eq. 1)".

**RC:** *Line 240 "Since areas are used to interpolate water heights..there is no need to derive an estimation of water area for the whole lake." Please elaborate on or clarify this sentence. If the H-A hypsometric curves are developed based on water body areas, how is it possible that the whole lake is not necessary for this assessment? Does this mean that half a lake can be used as long as it is always the same half of the lake (an underestimate of the total area) that is developed in the H-A curve?*

**AR:** This point was also raised by referee 1. Sentences with modifications (in red): "However, the method developed here is not very sensitive to systematic errors in water body surface area detection.  Given that water areas are only used to estimate water heights via the A-H curve, systematic errors in water area detection will not affect the final height estimation. For example, systematically missing a part of the lake in the water area detection (truncation) will modify the absolute water area values in the A-H curve but will not change the water height values. This situation is encountered for some lakes without a well identified connection to a river or with another lake or lakes that overflow, for example downstream of a dam.

"

Figure AR1, shows a case study that illustrates our point.

**RC:** *Line 253. I would suggest saying that the retracking algorithm was designed for an ice surface, reducing its applicability to this task, as noted previously by Cretaux et al. 2018. (Not calling it a failure of the algorithm).*

AR: OK, we agree. Original sentence: "The retracking algorithm failures (the retracking being designed for ice surface, Crétaux et al., 2018) or the prolonged drying of certain lakes for entire years also do not allow to have enough data to construct the hypsometric curve (ephemeral lakes or ponds)."
Modifications (in red): "Sometimes we do not have enough data to construct the A-H curve even for lakes below the track. For some cases this is due to the fact that certain lakes are dry for most part of the time span. For other few cases, the retracking algorithm, which was designed for ice surfaces (Crétaux et al., 2018), does not provide consistent water heights."

**RC:** *Is page 17 (Lines 273-304) a single paragraph?*
*It seems that this could be split into 2-3 paragraphs on water balance and ground water, and anthropogenic effects and the management of reservoirs*

AR: We split the paragraph into:
water losses./ The five-year averaged (line 277)
catchment./ The reservoirs (line 294)

**RC:** *Lines 259-261: Because the RMSE of the Cooley et al. 2021 study, and the RMSE of the altimeters used here are not included in the text, it is not possible to determine if the differences in the medians (2.06m this study; 1.60m Cooley et al. 2021), is significant. This idea is related to the earlier mentioned concept on line 39, where studies have asserted RMSEs of 0.69m. If that is the case then the differences in the medians between this study and Cooley et al 2021 may be within the margin of error.*

AR: We have included an evaluation of our results in the revised manuscript (see Supplement). The RMSE and bias were estimated for two lakes with available in situ V-H data. We found RMSE of 0.073 m and 0.015 m, and biases of –0.070 m and 0.006 m. This is smaller than the difference between the median height values provided by Cooley et al. (2021) and by our results.